# Suitability of resampled multispectral datasets for mapping flowering plants in the Kenyan savannah

David Masereti Makori[1,2]*, Elfatih M. Abdel-Rahman[1,3], Tobias Landmann[1], Onisimo Mutanga[2], John Odindi[2], Evelyn Nguku[1], Henry E. Tonnang[1], Suresh Raina[1]

**1** International Centre of Insect Physiology and Ecology (ICIPE), Nairobi, Kenya, **2** School of Agricultural, Earth and Environmental Sciences, University of KwaZulu-Natal, Pietermaritzburg, South Africa, **3** Department of Agronomy, Faculty of Agriculture, University of Khartoum, Khartoum North, Sudan

* davidmakori@gmail.com, dmakori@icipe.org

**Data Availability Statement:** All relevant data are within the paper and its Supporting Information files.

## Abstract

Pollination services and honeybee health in general are important in the African savannahs particularly to farmers who often rely on honeybee products as a supplementary source of income. Therefore, it is imperative to understand the floral cycle, abundance and spatial distribution of melliferous plants in the African savannah landscapes. Furthermore, placement of apiaries in the landscapes could benefit from information on spatiotemporal patterns of flowering plants, by optimising honeybees' foraging behaviours, which could improve apiary productivity. This study sought to assess the suitability of simulated multispectral data for mapping melliferous (flowering) plants in the African savannahs. Bi-temporal AISA Eagle hyperspectral images, resampled to four sensors (i.e. WorldView-2, RapidEye, Spot-6 and Sentinel-2) spatial and spectral resolutions, and a 10-cm ultra-high spatial resolution aerial imagery coinciding with onset and peak flowering periods were used in this study. Ground reference data was collected at the time of imagery capture. The advanced machine learning random forest (RF) classifier was used to map the flowering plants at a landscape scale and a classification accuracy validated using 30% independent test samples. The results showed that 93.33%, 69.43%, 67.52% and 82.18% accuracies could be achieved using WorldView-2, RapidEye, Spot-6 and Sentinel-2 data sets respectively, at the peak flowering period. Our study provides a basis for the development of operational and cost-effective approaches for mapping flowering plants in an African semiarid agroecological landscape. Specifically, such mapping approaches are valuable in providing timely and reliable advisory tools for guiding the implementation of beekeeping systems at a landscape scale.

## 1. Introduction

African savannahs are characterized by unreliable and erratic rainfall with low and dispersed forest pockets. They do not efficiently support rainfed agriculture, necessitating for alternative sources of income to supplement the unpredictable crop yield [1,2]. Apiculture and related

**Funding:** The European Union funded the project "African reference laboratory (with satellite stations) for the management of pollinator bee diseases and pests for food security" under grant number DCI-FOOD-2011/023-520. The European Union also facilitated the acquisition of the April 2014 WorldView-2 data that was used in this study. The University of Helsinki funded the research community GIMMEC–Geoinformatics for monitoring and modelling of environmental change for facilitating the acquisition and processing of the AISA Eagle hyperspectral data. DMM, EN, SR, and TL's salaries were made possible by the financial support given by the European Union through project DCI-FOOD-2011/023-520. EMAR, EN, HEZT, and TL received salaries from icipe. icipe core funding was provided by UK Aid from the Government of the United Kingdom; Department for International Development Cooperation Agency (Sida); the Swiss Agency for Development for Economic Cooperation and Development (BMZ), Germany Federal Democratic Republic of Ethiopia; and the Kenyan Government. The funders had no role in study design, data collection and analysis, decision to publish, or preparation of the manuscript.

**Competing interests:** The authors have declared that no competing interests exist.

ecosystem services such as pollinator activities boosts local economies, food and nutritional security and improve biodiversity, hence valuable and sustainable socio-ecological practices in the African savannah [3,4].

Beneficial insects such as honeybees, stingless bees, wasps, butterflies, mulberry and wild silk moths [2,5], are important income sources to local communities and living adjacent to the forest, paramount in pollination and pivotal incentives in forest conservation. These insects are valued for among others production of honey, wax, dyes and silk on one hand [6,7], and pollination of agricultural and forest ecosystems, as well as conservation of derelict land and degraded forest [2,8]. It is estimated that between 60% to 90% of plant species depend on insects and other animals such as birds for pollination [2,9]. In the USA for instance, pollination is estimated to contribute more than $14 billions a year to the agricultural economy alone [4,10]. However, whereas the contribution of pollination in Africa is unequivocal, its economic importance is yet to be fully documented.

As human pressure on land increases, communities living within four kilometers radius from the forest increases. These communities directly or indirectly depend on a range of forest resources and ecosystem services [3,4]. For instance, placement of apiaries in close proximity of a forest (less than one kilometer) doubles the production of honey than when placed out of a three kilometer radius [11]. Moreover, proximity to the right types and amounts of pollen and nectar improves hive productivity, honey quality and the agility of bees to fight off pests and diseases [5,11,12]. Forest habitats are important to the various life cycles of many beneficial insect species [13,14]. As aforementioned, these insects are useful for among others pollination, improving biodiversity, diversification of livelihood options and as natural agents of pest control [15,16]. Disturbance of forest stands and savannah vegetation leads to reduction in pollen and nectar sources and ultimately pollinators, which are susceptible to habitat alteration and changes in climate [17]. A decline or elimination of some plant species leads to a reduction of certain type of pollinators which are specific to either pollen type, flower colour and morphology or physiology [18]. Most insects rely on visual signals in the choice of flowers to visit. Colour, shape and size influence insects in flower preference [19–21]. For instance, hummingbirds prefer red coloured flowers, flies like pale colours, while butterflies and bees prefer brightly coloured flowers [19–23]. Since most of the crops are pollinated by social pollinators such as honeybees, agricultural production in Africa is predicted to reduce as their numbers decline [24,25]. In this regard, measures aimed at locating, conserving and improving cover of relevant flowering vegetation around vulnerable communities that depend on agriculture for their livelihoods are necessary.

Geoinformation and earth observation tools are increasingly being used to establish, locate and secure sources of nectar and pollen for enhanced hive productivity and ecosystem services [12,26,27]. Specifically, plant species mapping techniques adopting remotely sensed data assumes that each plant species has a unique spectral niche that is defined by the species biophysical and biochemical make-up [28–30]. Therefore, it is possible to identify and separate every tree species using their spectral features. However, commonly, operational mapping of tree species using remote sensing systems is hampered by the low spectral resolution in multispectral images and high acquisition cost of hyperspectral images. The improved division of the electromagnetic spectrum in hyperspectral data for instance gives narrow band data the ability to resolve subtle spectral canopy features associated with carotenoid, chlorophyll content and foliar nutrient content [29,31,32]. However, prohibitive cost, high dimensionality and multicollinearity, especially when using conventional parametric classification and regression procedures often make the use of hyperspectral data unfeasible. Specifically, the Hughes effect and the high redundancy rates of some bands in models developed using hyperspectral data impede landscape classification [33–36]. In this regard, it is paramount to explore the utility of

multispectral images with fewer broad bands for optimal discrimination of functional flowering groups [29,37–39]. However, whereas broadband multispectral data of lower spectral and medium spatial resolution such as the Landsat series have become popular in landscape mapping, they could mask out specific spectral features of functional flowering groups, resulting in very low mapping accuracy. The newly launched relatively improved spectral and/or spatial resolution sensors such as WorldView-2, RapidEye, Spot-6 and Sentinel-2 offer great potential in detecting different colours of functional flowering groups [40–42]. Such sensors are specifically designed to capture spectral properties at additional wavebands such as red-edge and yellow spectrum that mimic over 90% of plant biophysiological information [43–45]. However, the acquisition cost of some of the commercial multispectral data could limit their operational mapping applications. Hence, there is need first to explore the utility of simulated image data of such multispectral sensors and compare their usefulness with freely available ones for flower mapping [46,47].

In this study, we explored the utility of four simulated multispectral data (i.e. WorldView-2, RapidEye, Spot-6 and Sentinel-2) for detecting and mapping functional flowering groups in the African savannahs during the beginning and peak flowering seasons.

## 2. Methods

### 2.1 Study area

The study was carried in Kasanga (0.770˚S and 38.143˚E and approximately 933 metres above sea level), about 17 km north of Mwingi town, in Mwingi Central Sub-county, Kitui County, Kenya (Fig 1). The study area covers about 7.88 km² in a semi-arid agroecological zone with relatively high temperatures (ranging from 15˚C to 31˚C). The lowest temperatures are experienced between the months of July to August, while higher temperatures are experienced twice a year, from February to March and September to October. Rainfall in Mwingi is relatively low with typically two peaks in April and November (mean annual precipitation of between 147 to 270) [38]. Vegetation in Mwingi varies from woody plants, shrubs to crops. They include *Azandirachta indica*, *Melia volkensii*, *Markhamia lutea*, *Zizyphus abyssinica*, *Albizia gummifera* and *Acacia* spp. Flowering plants in Mwingi include *Terminalia brownie*, *Cassia diambotia*, *Aspilia mozambensis*, *Solonium incunum*, *Cassia semea*, *Grewia* spp, *Boscia* spp, and *Acacia* spp [48]. Most of these plants start flowering from December to May, with peak flowering season in February. The flowering season is triggered by the onset of rainfall in November, while the March-April rainfall extends the flowering through to May.

In Mwingi, traditional agricultural practices such as tilling, bush clearance and charcoal burning lead to deforestation of natural forest patches, including melliferous plants. This in turn leads to reduction in honeybee products, pollination services and biodiversity [38,49,50].

### 2.2 Image acquisition and pre-processing

**2.2.1 AISA Eagle hyperspectral images.** The airborne AISA Eagle hyperspectral images were obtained during the onset and peak of flowering periods, i.e. 11th January 2014 and 14 February 2013, respectively. AISA Eagle has a pushbroom scanning sensor with 0.037˚ instantaneous field of view and 36.04˚ and 969 pixels across the spatial axis [53]. To produce an optimal number of bands with high signal-to-noise-ratio (SNR), the sensor was set on eight times spectral binning at full width at half maximum (FWHM) of 8–10.5 nm in the 400 to 1000 nm spectral range. Hence, the product had 64 bands and a 0.6m spatial resolution after georeferencing. Since this study sought to re-sample AISA Eagle hyperspectral images to a range of multispectral sensor specifications, all AISA Eagle image bands were utilized to accommodate a range of sensor characteristics.

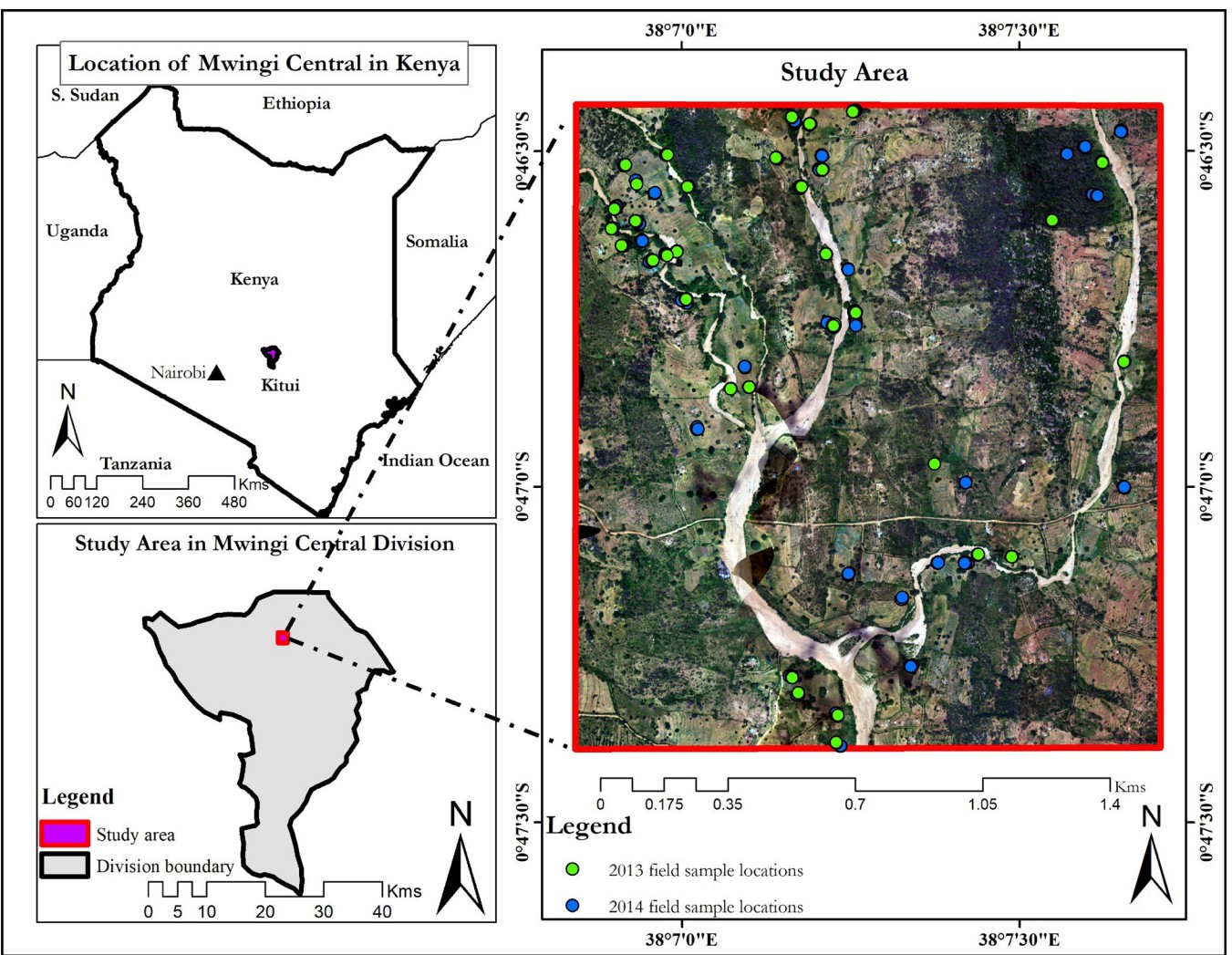

**Fig 1. The study area indicating the field data sample points collected from the study site in Mwingi in January 2014 and February 2013.** The image on the background is a true colour composite AISA Eagle captured in February 2013 over the Mwingi study site. The country boundary data was downloaded from the World Resource Institute website (https://www.wri.org/resources/data-sets/kenya-gis-data) [51]. The maps were developed using QGIS software version 3.10 (https://qgis.org/en/site/) [52].

A digital elevation model (DEM) with 90m resolution, interpolated to match the 0.6m resolution of the images [54], together with a 2m WorldView-2 image (37S UTM projection) captured over Mwingi in April 2014 were used to geo-reference the AISA Eagle images. The AISA Eagle raw digital values were also converted to at-sensor spectral radiance using the CaliGeo-Pro atmospheric correction tool (Specim Limited, Oulu, Finland).

**2.2.2 Resampling of multispectral data from AISA Eagle.** Both AISA Eagle hyperspectral images captured in January 2014 and February 2013 were resampled to WorldView-2, RapidEye, Spot-6 and Sentinel-2 multispectral spatial and spectral sensor specifications using the spectral resampling tool in Environment for Visualizing Images (ENVI) 5.3 software [55]. ENVI uses the full-width-half-maximum wavelength information to spectrally resample images. Since the spectral responses of these multispectral images were predefined in ENVI, the pre-defined filter function was used as a resampling method under the spectral resampling tool. These four multispectral image data were used to test the possibility of mapping flowering

plants using 10 m or less spatial resolution. Also, we selected multispectral sensors that have infrared and red edge bands that have been established to be effective for detecting vegetation and flowers spectral signals [37,56]. Resampled multispectral data were used to avoid image acquisition cost before testing their suitability for mapping flowering plants. Further, the Sentinel-2 sensor was not yet launched when the study was conducted. Therefore, we obtained AISA Eagle data from previous studies [37,38] that looked at the possibility of mapping flowering plants using hyperspectral image data.

Four multispectral sensors (WorldView-2, RapidEye, Spot-6 and Sentinel-2) were considered for this study. These images were resampled from their respective January 2014 and February 2013 AISA Eagle images. Additionally, a WorldView-2 image captured over the study site in April 2014 was used to compare the accuracy of the classification achieved using the resampled WorldView-2 image and provide insight on the reliability for our image resampling procedure.

*2.2.2.1 WorldView-2.* WorldVeiw-2 is an eight (8) waveband multispectral image sensor that was launched in October 2009 with a 1.84 m spatial resolution and an additional 0.46 metre panchromatic band [42,57,58]. The WorldView-2 sensor operates at an altitude of 770 kilometres and images at coastal, blue, green, yellow, red, red edge, near infrared 1, and near infrared 2 regions of the electromagnetic spectrum (EMS). These wavebands have distinct spectral separation (Table 1), hence potential for differentiating vegetation communities, health and other spectral features [29,33,42,58]. The cost of acquiring a tasked WorldView-2 image is $58 per km$^2$, with a 100 square kilometre and 5-kilometre minimum order width.

*2.2.2.2 RapidEye.* RapidEye is a 5-waveband multispectral sensor with 5 m spatial resolution (Table 2) that was launched in August 2008 [59,60]. In addition to other multispectral bands, RapidEye is notable for its red edge band which has potential for mapping flower blooms [37]. The cost of RapidEye image is $1.90 per square kilometre, with a 500 square kilometre and 10-kilometre minimum order width.

*2.2.2.3 Spot-6.* Spot-6 is a multispectral sensor launched in September 2012. It is characterised by four wavebands with 6 m spatial resolution and a panchromatic band with 1.5 metres spatial resolution [61]. Table 3 further details the sensor spectral and spatial information. The cost of Spot-6 image is $5.75 per square kilometre, with a 500 square kilometre and 20-kilometre order width.

*2.2.2.4 Sentinel-2.* Sentinel-2 is multispectral sensor that was launched in 2015. It is characterised by 13 bands in the spectral ranges of visible/near infrared (VNIR) and shortwave infrared (SWIR) (Table 4), with spatial resolution ranging from 10 to 60 metres [40,62,63]. Sentinel-2 data are freely available. When this study was conducted, the sensor was not yet launched hence the need to resample this dataset from AISA Eagle images.

**Table 1. Spectral responses of WorldView-2 indicating the lower and upper wavelength with the specific resolution of each band.**

|  | Waveband | Lower λ (nm) | Upper λ (nm) | Spatial resolution (m) |
|---|---|---|---|---|
| 1 | Panchromatic | 450 | 800 | 0.46 |
| 2 | Coastal | 400 | 450 | 1.84 |
| 3 | Blue | 450 | 510 | 1.84 |
| 4 | Green | 510 | 580 | 1.84 |
| 5 | Yellow | 585 | 625 | 1.84 |
| 6 | Red | 630 | 690 | 1.84 |
| 7 | Red edge | 705 | 745 | 1.84 |
| 8 | Near infrared 1 | 770 | 895 | 1.84 |
| 9 | Near infrared 2 | 860 | 1040 | 1.84 |

**Table 2. RapidEye wavebands indicating the lower and centre wavelength with the specific resolution of each band.**

| | Waveband | Lower λ (nm) | Centre λ (nm) | Spatial resolution (m) |
|---|---|---|---|---|
| 1 | Blue | 440 | 475 | 5 |
| 2 | Green | 520 | 555 | 5 |
| 3 | Red | 630 | 657.5 | 5 |
| 4 | Red edge | 690 | 710 | 5 |
| 5 | Near infrared | 760 | 805 | 5 |

**2.2.3 Field data collection.** A stratified random sampling approach was used to collect field data on white and yellow flowering plants, flowering fobs, shrubs, green trees, senesced trees and crops (maize and sorghum) within three days of the AISA Eagle and WorldView-2 image data acquisition. Data on crops were only collected in January 2014 as they had already been harvested in February 2013. Additionally, trees that had chlorophyll-inactive leaves were identified, collected and tagged as 'senesced trees'. The Geospatial Modelling Environment (GME) tool was used to randomly generate 156 ground control points (GCPs) within the study site (Fig 1), from which the field data were collected. Table 5 details the number of samples on each sensor data, depending on their spatial resolution (pixel size) during the flowering seasons (i.e. January 2014; onset of flowering, February 2013; peak flowering and April 2014; end of flowering). The variations in the number of samples on each image data is due to differences in sensor pixel sizes.

To locate the randomly generated GCPs, the data were loaded into a global positioning system (GPS) device with a 3-metre accuracy that was used to locate identified trees on the ground. Trees with three (3) metre canopy sizes (or more) were tagged and measurements taken to ensure all GCP readings were within the tree crowns. Other GCP's were collected from a 10 cm spatial resolution Nikon D3X digital camera image that was captured together with the AISA Eagle image data. Photos of the various functional groups are presented in S1 Table.

**2.2.4 Random forest classifier.** The supervised random forest (RF) ensemble algorithm [64–66] with recursive partitioning was used to classify the different multispectral datasets. The ensemble is a robust machine learning algorithm that allows for growing of many regression trees (*ntree*) from bootstrap samples with replacement from the original data. It uses a majority voting procedure to assign classes to the reference datasets. Each tree uses two thirds (67%) of the randomly and independently selected dataset (*mtry*) for training the algorithm and one third (33%) of the remaining dataset for testing its accuracy [64] using the out-of-bag (OOB) instances. All the bands in the resampled multispectral images were used as variables in the prediction while being optimized on the OOB error rate [66] using grid search and a 10-fold cross-validation method [67]. RF has a high level of randomness in bagging (selection of datasets) with low sensitivity towards noise and overtraining, making it more suitable for

**Table 3. Spot-6 wavebands showing the lower, upper and centre wavelength with the resolution of specific bands that make up the image.**

| | Waveband | Lower λ (nm) | Centre λ (nm) | Spatial resolution (m) |
|---|---|---|---|---|
| 1 | Panchromatic | 450 | 597.5 | 1.5 |
| 2 | Blue | 450 | 487.5 | 6 |
| 3 | Green | 530 | 560 | 6 |
| 4 | Red | 625 | 660 | 6 |
| 5 | NIR | 760 | 825 | 6 |

**Table 4. Sentinel-2 wavebands showing the central wavelength, bandwidth with the resolution of specific bands that make up the image.**

| | Waveband | Central λ (nm) | Bandwidth (nm) | Spatial resolution (m) |
|---|---|---|---|---|
| 1 | Coastal aerosol | 442.7 | 21 | 60 |
| 2 | Blue | 492.4 | 66 | 10 |
| 3 | Green | 559.8 | 36 | 10 |
| 4 | Red | 664.6 | 31 | 10 |
| 5 | Vegetation red edge | 704.1 | 15 | 20 |
| 6 | Vegetation red edge | 740.5 | 15 | 20 |
| 7 | Vegetation red edge | 782.8 | 20 | 20 |
| 8 | Near infrared | 832.8 | 106 | 10 |
| 8A | Narrow near infrared | 864.7 | 21 | 20 |
| 9 | Water vapour | 945.1 | 20 | 60 |
| 10 | Shortwave infrared–Cirrus | 1373.5 | 31 | 60 |
| 11 | Shortwave infrared | 1613.7 | 91 | 20 |
| 12 | Shortwave infrared | 2202.4 | 175 | 20 |

simultaneous classification and variable selection [64,66,68]. The randomForest [64] library in 'R statistical software' version 3.6.1 [66] was used for this study.

**2.2.5 Accuracy assessment.** An independent 30% random sample of the reference data points was used to test the classification accuracy of the RF classification model. To establish the versatility of the models, the following matrices were calculated; overall accuracy (OA), users' accuracy (UA), producer's accuracy (PA), quantity disagreement (QD) and allocation disagreement (AD) [69–71]. QD and AD are absolute values that are used to evaluate the difference among predictions and reference data while comparing the percentage of observations that do not have optimal spatial locations as opposed to the reference samples [71]. It gives a better indication of the portion of the predictions that are of a good fit with the prediction data [72,73].

**Table 5. Field reference data used in the classification of the various flowering and other plant classes from different image datasets.** 70% of the data were used for training while 30% were used to test the accuracy of the random forest classification models.

| Class | AISA Eagle | WorldView-2 | RapidEye | Spot-6 | Sentinel-2 | |
|---|---|---|---|---|---|---|
| | | | February 2013 | | | |
| Flowering fobs | 320 | 89 | 115 | 34 | 31 | |
| Green trees | 241 | 36 | 54 | 21 | 15 | |
| Senesced trees | 450 | 85 | 211 | 66 | 56 | |
| Shrubs | 639 | 109 | 149 | 53 | 38 | |
| Soil | 1,369 | 432 | 422 | 142 | 111 | |
| White flowers | 472 | 84 | 161 | 54 | 44 | |
| Yellow flowers | 319 | 77 | 184 | 53 | 51 | |
| | | | January 2014 | | | WorldView-2 (April 2014) |
| Flowering fobs | 1,659 | 155 | 22 | 27 | 35 | 148 |
| Green trees | 2,503 | 226 | 36 | 52 | 44 | 224 |
| Senesced trees | 1,178 | 107 | 19 | 43 | 13 | 101 |
| Shrubs | 4,304 | 392 | 65 | 112 | 83 | 389 |
| Soil | 8,740 | 787 | 126 | 236 | 161 | 783 |
| White flowers | 980 | 93 | 18 | 41 | 14 | 87 |
| Yellow flowers | 893 | 85 | 15 | 42 | 16 | 76 |

## 3. Results

### 3.1 Flower compaction and spread

The classification maps for the different flowering seasons; onset of flowering season (January 2014), maximum flowering season (February 2013) and end of the flowering season (April 2014), are presented in Figs 2–4, respectively.

Generally, flower classes were mapped with lower individual class accuracies (up to 87.36%) during the onset of the flowering season (Fig 2 and Table 6) as opposed to the maximum flowering season (up to 93.33%, Fig 3 and Table 6), using different multispectral sensor data. Specifically, both yellow and white flowers were reliably delineated with accuracies of up to 94.44% and 90.00% respectively (Table 6). These accuracies were higher than mapping accuracies of other non-flowering plants (Table 6). There was more flower compaction in February (peak flowering season) than January (onset of flowering season). Furthermore, there was more flowering fobs spread across the Mwingi study site in the peak flowering season compared to the onset of the flowering season, but less towards the end of the flowering season (Fig 4 and Table 7). Flowering fobs had the highest classification accuracy compared to the other flower classes (Table 6). The spread of the flowering fobs reduced as the peak flowering season came to an end towards April (Tables 6 and 7). Typically, April is the end flowering season in Mwingi, hence the reduced flowering. On the other hand, flowering crops reduced towards the peak flowering period as flowering in other melliferous plants increased. Results also show that crops were mapped with relatively higher accuracy (79.31%) at the start than later in the flowering season (65.08%).

The delineation of flowering plant species differed with sensor spatial and spectral resolutions, with a decrease in spatial and spectral resolution, leading to a decline in classification accuracies (Tables S1 and 6 and 7). WorldView-2 generated the best overall classification results (87.36% at the onset, 93.33% at the peak and 76.40% at the end of the flowering seasons, Tables S1 and 6). Using the WorldView-2 data, subtle flowering differences were identified between the three capture periods; i.e. January 2014, February 2013 and April 2014.

Visual observation of the classification maps indicated that Sentinel-2 had the poorest presentation of the different flowering plants under consideration (Figs 2 and 3). The image and classes were more pixilated with more 'salt and pepper' than classification maps from other sensors. However, the mapping accuracy presented in Tables S1 and 7 indicated that Sentinel-2 data mapped the flowering plants more accurately than RapidEye and Spot-6 data, even though the latter two had better spatial resolution. Spectrally, RapidEye, Spot-6, WorldView-2 and Sentinel-2 have five, six, eight and twelve spectral bands respectively, a clear indication that spectral resolution influence mapping of melliferous plant species. On the other hand, our maps show over prediction of senesced tree class, particularly during the peak flowering season (February 2013).

### 3.2 Accuracy assessment

Results presented in Tables S1 and 7 showed consistent differences between the January 2014 and February 2013 classification maps and between the different sensor datasets. WorldView-2 data had superior classification accuracy (87.36% in January 2014 and 93.33% in February 2013) than all the other sensor datasets. Spot-6 had the least overall accuracy (67.52%) in February 2013 while RapidEye had the least overall accuracy (72.84%) in January 2014. Both the users' and producers' accuracies reduced as the spatial resolution decreased (Table 6). On the other hand, the users' and producers' accuracies of Sentinel-2 were higher than those of both RapidEye and Spot-6.

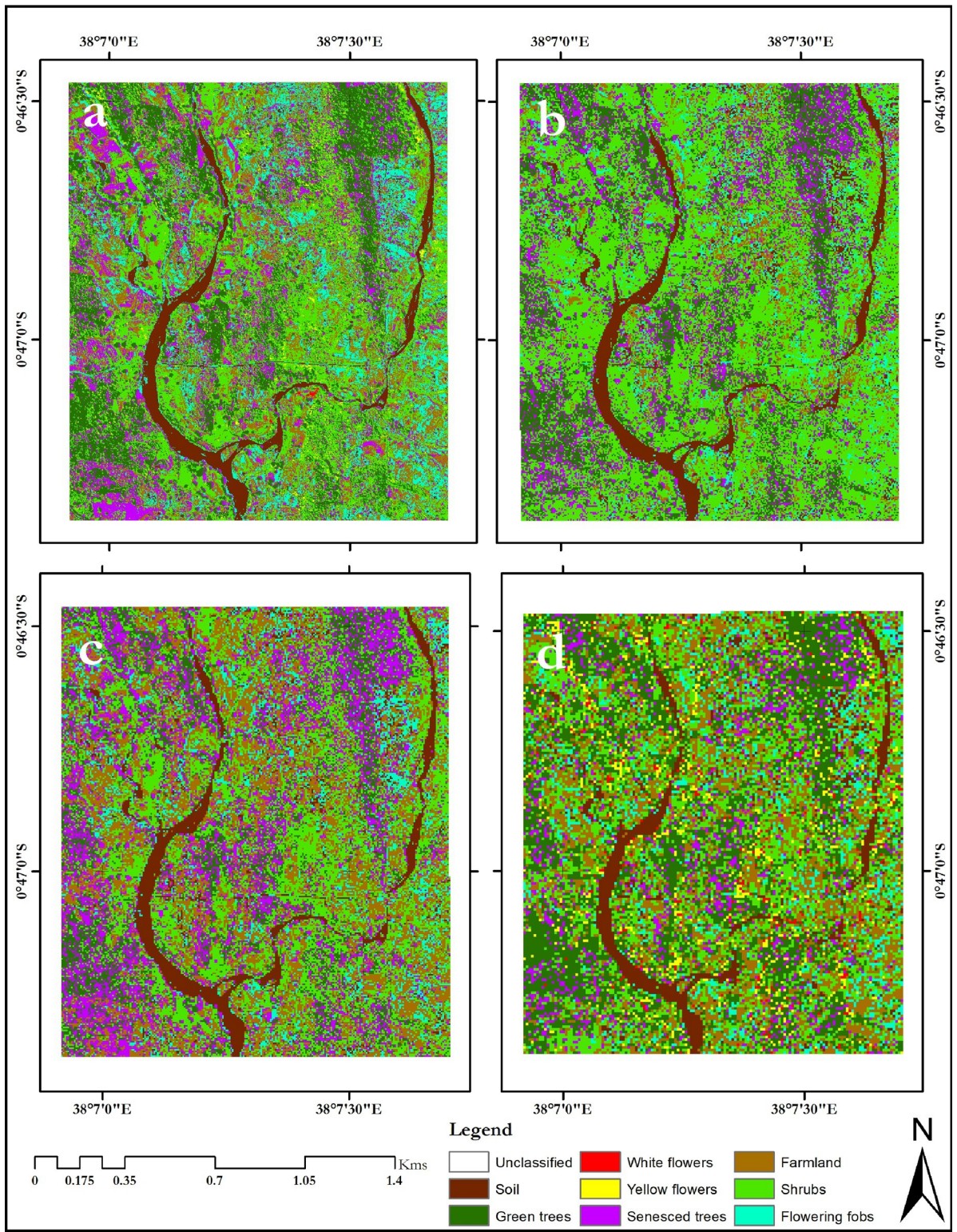

**Fig 2.** January 2014 classification maps of the Mwingi study site obtained using random forest classifier and resampled simulated WorldView-2 image classification (a), RapidEye (b), Spot-6 (c), Sentinel-2 (d) images. The maps were developed using QGIS software version 3.10 (*https://qgis.org/en/site/*) [52].

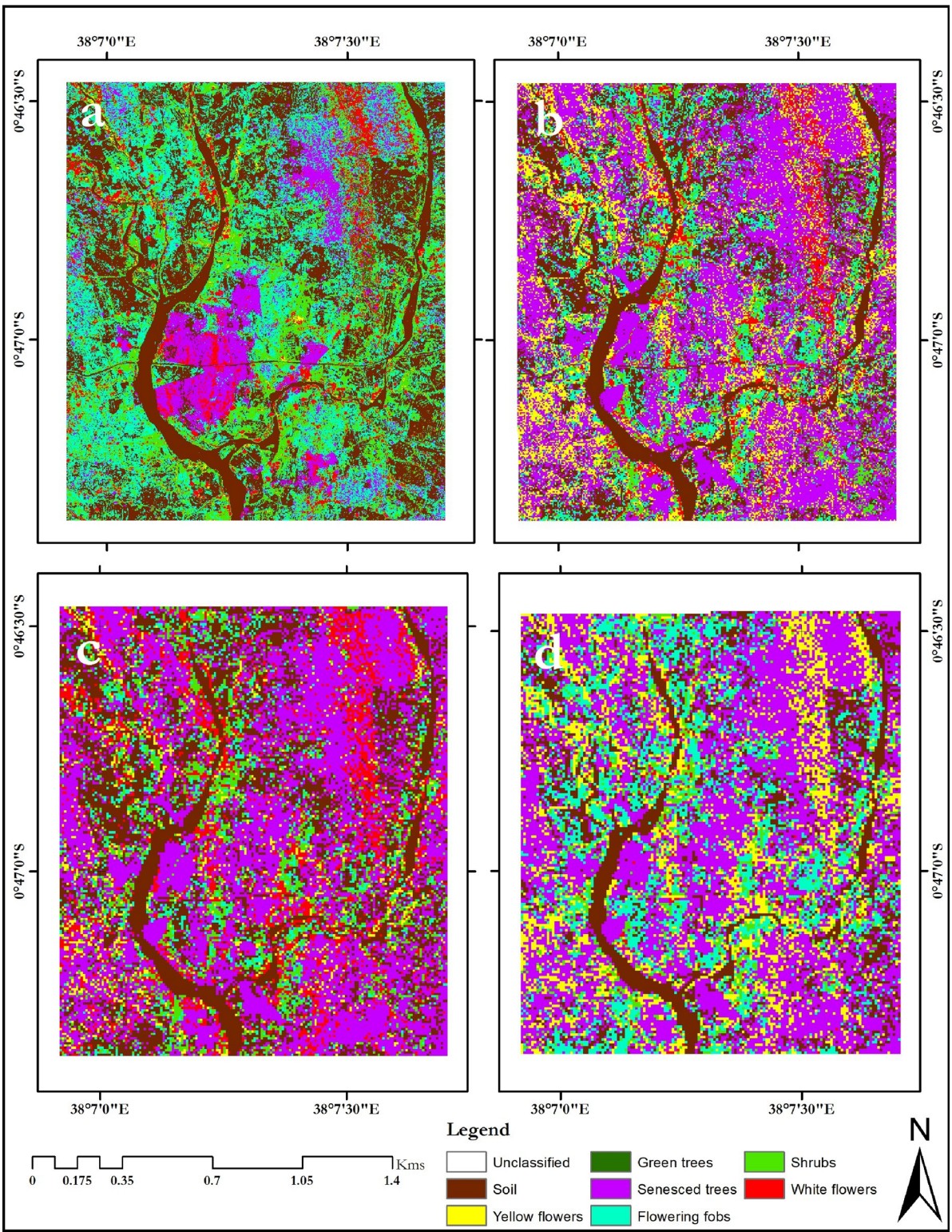

**Fig 3.** February 2013 classification maps of the Mwingi study site obtained using random forest classifier and resampled simulated WorldView-2 image classification (a), RapidEye (b), Spot-6 (c), Sentinel-2 (d) images. The maps were developed using QGIS software version 3.10 (*https://qgis.org/en/site/*) [52].

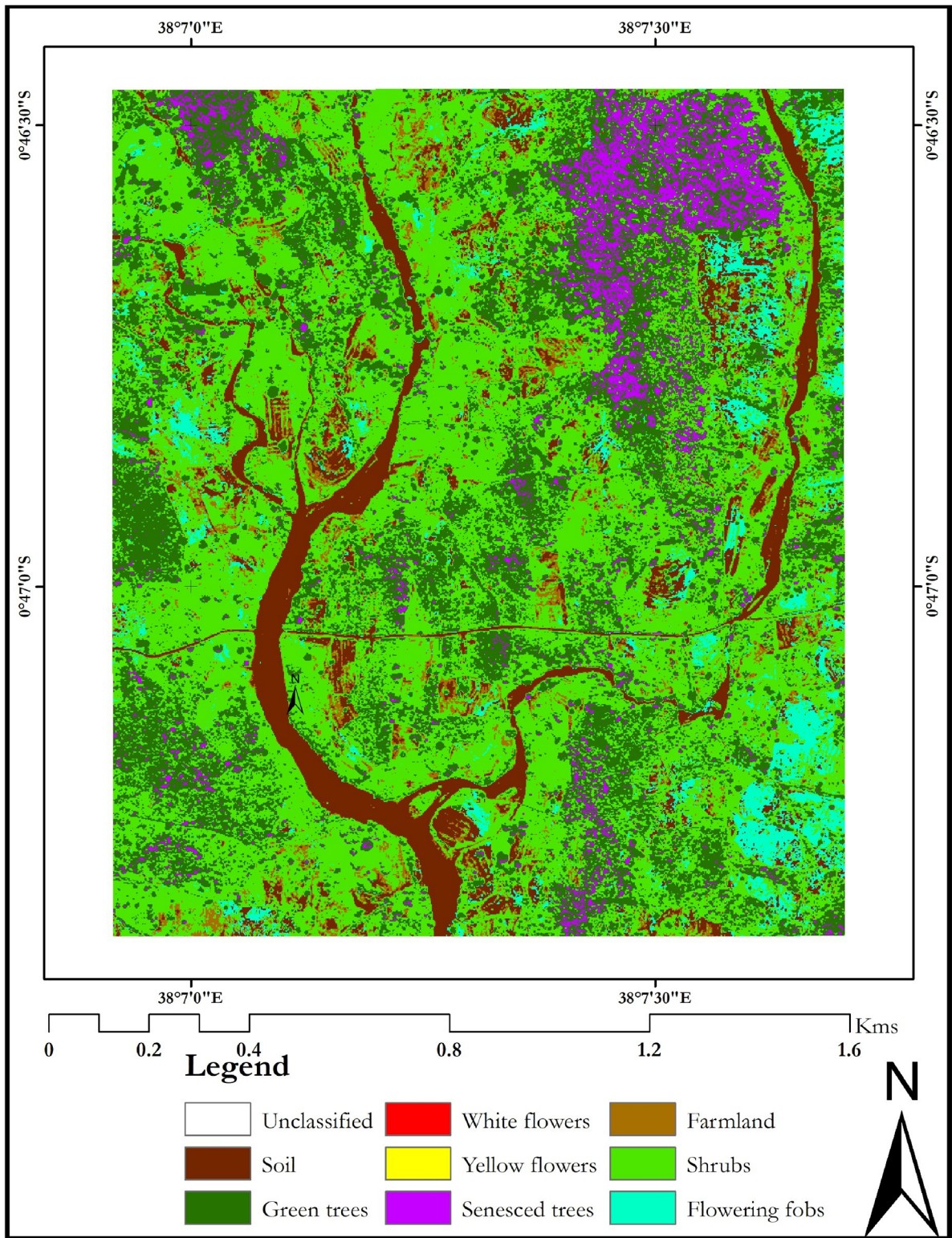

**Fig 4. April 2014 classification map of the Mwingi study site obtained using random forest classifier and WorldView-2 image.** The map was developed using QGIS software version 3.10 (*https://qgis.org/en/site/*) [52].

**Table 6. Confusion matrix for the classification of flowering vegetation communities in the Mwingi study site obtained using random forest classifier and resampled simulated WorldView-2, RapidEye, Spot-6 and Sentinel-2 datasets for onset of flowering (January 2014) and maximum flowering (February 2013).**

| | January 2014 | | | | | | | | | February 2013 | | | | | | | | |
|---|---|---|---|---|---|---|---|---|---|---|---|---|---|---|---|---|---|---|
| | FB | GT | ST | SB | SL | WF | YF | Totals | UA | FB | GT | ST | SB | SL | WF | YF | Totals | UA |
| | | | | | | | | | WorldView-2 | | | | | | | | | |
| FB | 38 | 0 | 0 | 3 | 2 | 0 | 0 | 43 | 88.4 | 25 | 0 | 1 | 1 | 0 | 1 | 0 | 28 | 89.3 |
| GT | 0 | 62 | 1 | 8 | 0 | 2 | 0 | 73 | 84.9 | 0 | 10 | 0 | 0 | 0 | 1 | 0 | 11 | 90.9 |
| ST | 0 | 0 | 24 | 1 | 0 | 0 | 0 | 25 | 96 | 1 | 0 | 24 | 0 | 0 | 0 | 0 | 25 | 96 |
| SB | 3 | 5 | 5 | 92 | 0 | 15 | 14 | 134 | 68.7 | 0 | 0 | 0 | 29 | 0 | 5 | 5 | 39 | 74.4 |
| SL | 2 | 0 | 0 | 0 | 234 | 0 | 0 | 236 | 99.2 | 0 | 0 | 0 | 0 | 129 | 0 | 0 | 129 | 100 |
| WF | 0 | 0 | 1 | 2 | 0 | 10 | 1 | 14 | 71.4 | 0 | 0 | 0 | 1 | 0 | 18 | 1 | 20 | 90 |
| YF | 0 | 0 | 1 | 2 | 0 | 0 | 10 | 13 | 76.9 | 0 | 0 | 0 | 1 | 0 | 0 | 17 | 18 | 94.4 |
| Totals | 43 | 67 | 32 | 108 | 236 | 27 | 25 | 538 | | 26 | 10 | 25 | 32 | 129 | 25 | 23 | 270 | |
| PA (%) | 88.4 | 92.5 | 75 | 85.2 | 99.2 | 37 | 40 | | | 96.2 | 100 | 96 | 90.6 | 100 | 72 | 73.9 | | |
| OA (%) | | | | | | | | | 87.4 | | | | | | | | | 93.3 |
| Kappa | | | | | | | | | 0.811 | | | | | | | | | 0.908 |
| AD (%) | | | | | | | | | 6.69 | | | | | | | | | 2.96 |
| QD (%) | | | | | | | | | 5.95 | | | | | | | | | 3.70 |
| | | | | | | | | | RapidEye | | | | | | | | | |
| FB | 3 | 0 | 0 | 1 | 0 | 0 | 0 | 4 | 75 | 16 | 2 | 2 | 4 | 0 | 2 | 4 | 30 | 53.3 |
| GT | 0 | 6 | 2 | 3 | 0 | 1 | 2 | 14 | 42.9 | 0 | 4 | 0 | 0 | 0 | 0 | 1 | 5 | 80 |
| ST | 0 | 1 | 3 | 1 | 0 | 1 | 0 | 6 | 50 | 7 | 1 | 48 | 3 | 4 | 2 | 3 | 68 | 70.6 |
| SB | 2 | 2 | 0 | 10 | 0 | 2 | 2 | 18 | 55.6 | 5 | 2 | 6 | 25 | 0 | 7 | 4 | 49 | 51 |
| SL | 0 | 0 | 0 | 0 | 37 | 0 | 0 | 37 | 100 | 2 | 0 | 4 | 0 | 122 | 2 | 1 | 131 | 93.1 |
| WF | 0 | 0 | 0 | 0 | 0 | 0 | 0 | 0 | 0 | 2 | 4 | 0 | 6 | 0 | 23 | 12 | 47 | 48.9 |
| YF | 0 | 1 | 0 | 0 | 0 | 1 | 0 | 2 | 0 | 2 | 3 | 3 | 6 | 0 | 12 | 30 | 56 | 53.6 |
| Totals | 5 | 10 | 5 | 15 | 37 | 5 | 4 | 81 | | 34 | 16 | 63 | 44 | 126 | 48 | 55 | 386 | |
| PA (%) | 60 | 60 | 60 | 66.7 | 100 | 0 | 0 | | | 47.1 | 25 | 76.2 | 56.8 | 96.8 | 47.9 | 54.6 | | |
| OA (%) | | | | | | | | | 72.8 | | | | | | | | | 69.4 |
| Kappa | | | | | | | | | 0.551 | | | | | | | | | 0.619 |
| AD (%) | | | | | | | | | 17.28 | | | | | | | | | 26.42 |
| QD (%) | | | | | | | | | 9.88 | | | | | | | | | 4.15 |
| | | | | | | | | | Spot-6 | | | | | | | | | |
| FB | 2 | 0 | 0 | 1 | 2 | 0 | 0 | 5 | 40 | 9 | 0 | 2 | 4 | 0 | 2 | 3 | 20 | 45 |
| GT | 0 | 9 | 1 | 2 | 0 | 1 | 2 | 15 | 60 | 0 | 4 | 0 | 0 | 0 | 3 | 3 | 10 | 40 |
| ST | 0 | 3 | 4 | 0 | 0 | 2 | 3 | 12 | 33.3 | 3 | 2 | 40 | 3 | 4 | 0 | 3 | 55 | 72.7 |
| SB | 0 | 1 | 3 | 24 | 0 | 4 | 3 | 35 | 68.6 | 6 | 0 | 1 | 19 | 1 | 9 | 4 | 40 | 47.5 |
| SL | 2 | 0 | 0 | 0 | 68 | 0 | 0 | 70 | 97.1 | 2 | 0 | 3 | 1 | 83 | 1 | 3 | 93 | 89.3 |
| WF | 0 | 1 | 0 | 2 | 0 | 2 | 0 | 5 | 40 | 3 | 3 | 1 | 3 | 0 | 15 | 7 | 32 | 46.9 |
| YF | 0 | 1 | 0 | 1 | 0 | 1 | 2 | 5 | 40 | 0 | 3 | 0 | 0 | 0 | 6 | 15 | 24 | 62.5 |
| Totals | 4 | 15 | 8 | 30 | 70 | 10 | 10 | 147 | | 23 | 12 | 47 | 30 | 88 | 36 | 38 | 274 | |
| PA (%) | 50 | 60 | 50 | 80 | 97.1 | 20 | 20 | | | 39.1 | 33.3 | 85.1 | 63.3 | 94.3 | 41.7 | 39.5 | | |
| OA (%) | | | | | | | | | 75.5 | | | | | | | | | 67.5 |
| Kappa | | | | | | | | | 0.620 | | | | | | | | | 0.597 |
| AD (%) | | | | | | | | | 17.69 | | | | | | | | | 24.09 |
| QD (%) | | | | | | | | | 6.80 | | | | | | | | | 8.39 |
| | | | | | | | | | Sentinel-2 | | | | | | | | | |
| FB | 5 | 0 | 0 | 2 | 0 | 0 | 0 | 7 | 71.4 | 6 | 0 | 0 | 1 | 0 | 0 | 0 | 7 | 85.7 |
| GT | 0 | 11 | 1 | 0 | 0 | 1 | 0 | 13 | 84.6 | 0 | 3 | 0 | 0 | 0 | 0 | 0 | 3 | 100 |

*(Continued)*

**Table 6.** (Continued)

| | January 2014 | | | | | | | | | February 2013 | | | | | | | | |
|---|---|---|---|---|---|---|---|---|---|---|---|---|---|---|---|---|---|---|
| ST | 0 | 1 | 2 | 0 | 0 | 0 | 0 | 3 | 66.7 | 0 | 0 | 14 | 0 | 0 | 0 | 0 | 14 | 100 |
| SB | 5 | 1 | 0 | 19 | 1 | 2 | 1 | 29 | 65.5 | 1 | 0 | 0 | 8 | 0 | 0 | 2 | 11 | 72.7 |
| SL | 0 | 0 | 0 | 1 | 47 | 0 | 0 | 48 | 97.9 | 0 | 0 | 2 | 0 | 33 | 0 | 1 | 36 | 91.7 |
| WF | 0 | 0 | 0 | 0 | 0 | 0 | 1 | 1 | 0 | 0 | 1 | 0 | 1 | 0 | 8 | 1 | 11 | 72.7 |
| YF | 0 | 0 | 0 | 0 | 0 | 1 | 2 | 3 | 66.7 | 2 | 0 | 0 | 1 | 0 | 5 | 11 | 19 | 57.9 |
| Totals | 10 | 13 | 3 | 22 | 48 | 4 | 4 | 104 | | 9 | 4 | 16 | 11 | 33 | 13 | 15 | 101 | |
| PA (%) | 50 | 84.6 | 66.7 | 86.4 | 97.9 | 0 | 50 | | | 66.7 | 75 | 87.5 | 72.7 | 100 | 61.5 | 73.3 | | |
| OA (%) | | | | | | | | 82.7 | | | | | | | | | 82.2 | |
| Kappa | | | | | | | | 0.699 | | | | | | | | | 0.777 | |
| AD (%) | | | | | | | | 10.58 | | | | | | | | | 10.89 | |
| QD (%) | | | | | | | | 6.73 | | | | | | | | | 6.93 | |

The overall, producers' and users' accuracies, Kappa, allocation disagreement (AD) and quantity disagreement (QD) are also presented.

Key:

FB - Flowering fobs

GT – Green trees

ST - Senesced trees

SB - Shrubs

SL - Soil

WF - White flowers

YF - Yellow flowers

PA - Predictor accuracy

UA - User accuracy

OA - Overall accuracy

AD - Allocation disagreement

QD - Quantity disagreement

Moreover, the maps had low quantity disagreement (QD) (3.70 to 9.88%) compared to the allocation disagreement (AD) (2.96 to 26.42%). Generally, there was more disagreement both in allocation and quantity as the spatial resolution decreased (Table 8). WorldView-2 had the best agreement fit in the two dimensions (as low as 2.96% AD), while RapidEye and Spot 6 had the highest disagreements (as high as 26.42% and 24.09% AD, respectively). On the contrary, Sentinel-2 with the least spatial resolution had better agreement (6.73% QD) than RapidEye and Spot-6.

Fig 5 compares deviation in means between different classes in the two flowering periods. Results showed that flowering plants had less deviation from the mean than non-flowering plants. There was lower deviation at the peak flowering season i.e. February 2013 than at the onset of flowering i.e. January 2014. Contrary to the others, yellow flowers and flowering fobs signatures deviated more from the mean during the peak flowering period than at the onset of flowering. In the study site, yellow flowers start flowering at the onset of October rains while other flowering plants have their peak in February. Generally, flowering fobs were confused with soil and other background features in January 2014 than February 2013.

## 4. Discussion

Results in this study showed that resampled multispectral data of different spectral and spatial resolutions can be used to reliably map flowering plants in a heterogenous landscape in the African Savanna at different flowering seasons. Multispectral images provide affordable or

**Table 7. Confusion matrix for the classification of flowering vegetation communities in the Mwingi study site obtained using random forest classifier and World-View-2 image for the end of the flowering season (April 2014) using the WorldView-2 image.**

| | | | | | Ground truth | | | | |
|---|---|---|---|---|---|---|---|---|---|
| Prediction | FB | GT | ST | SB | SL | WF | YF | Total | UA (%) |
| FB | 23 | 2 | 0 | 10 | 2 | 3 | 0 | 40 | 57.5 |
| GT | 1 | 40 | 12 | 6 | 0 | 6 | 8 | 73 | 54.8 |
| ST | 0 | 7 | 16 | 0 | 0 | 3 | 0 | 26 | 61.5 |
| SB | 8 | 13 | 0 | 83 | 6 | 7 | 7 | 124 | 66.9 |
| SL | 3 | 1 | 0 | 4 | 224 | 1 | 0 | 233 | 96.1 |
| WF | 1 | 1 | 2 | 2 | 0 | 5 | 2 | 13 | 38.5 |
| YF | 0 | 2 | 0 | 1 | 0 | 1 | 4 | 8 | 50 |
| Totals | 36 | 66 | 30 | 106 | 232 | 26 | 21 | 517 | |
| PA (%) | 63.9 | 60.6 | 53.3 | 78.3 | 96.6 | 19.2 | 19.1 | | |
| OA (%) | | | | | | | | | 76.4 |
| Kappa | | | | | | | | | 0.695 |
| AD (%) | | | | | | | | | 17.79 |
| QD (%) | | | | | | | | | 5.80 |

The overall, producers' and users' accuracies, Kappa, allocation disagreement (AD) and quantity disagreement (QD) are also presented.

Key:

FB – Flowering fobs

GT – Green trees

ST - Senesced trees

SB - Shrubs

SL - Soil

WF - White flowers

YF - Yellow flowers

PA - Producers' accuracy

UA - Users' accuracy

OA - Overall accuracy

AD - Allocation disagreement

QD - Quantity disagreement

freely available data useful for generating flower maps. Such maps could be operationally valuable in guiding the placement of honeybee apiaries at a landscape scale. Plants of different flower colour (i.e. white and yellow) were mapped with relatively higher accuracies. Specifically, we grouped the flowering plants in the study site into different functional flowering groups of different colours as honeybees foraging behaviour is highly influenced by among

**Table 8. Classification accuracy of different flowering vegetation communities in the Mwingi study site during the start of the flowering season (January 2014) and the peak of the flowering season (February 2013) using resampled simulated data.**

| Sensor | Overall Accuracy | | Allocation Disagreement | | Quantity Disagreement | |
|---|---|---|---|---|---|---|
| | 2014 | 2013 | 2014 | 2013 | 2014 | 2013 |
| WorldView-2 | 87.36 | 93.33 | 6.69 | 2.96 | 5.95 | 3.70 |
| WorldView-2 (April 2014) | 76.40 | N/A | 17.79 | N/A | 5.80 | N/A |
| RapidEye | 72.84 | 69.43 | 17.28 | 26.42 | 9.88 | 4.15 |
| Spot-6 | 75.51 | 67.52 | 17.69 | 24.09 | 6.80 | 8.39 |
| Sentinel-2 | 82.69 | 82.18 | 10.58 | 10.89 | 6.73 | 6.93 |

 

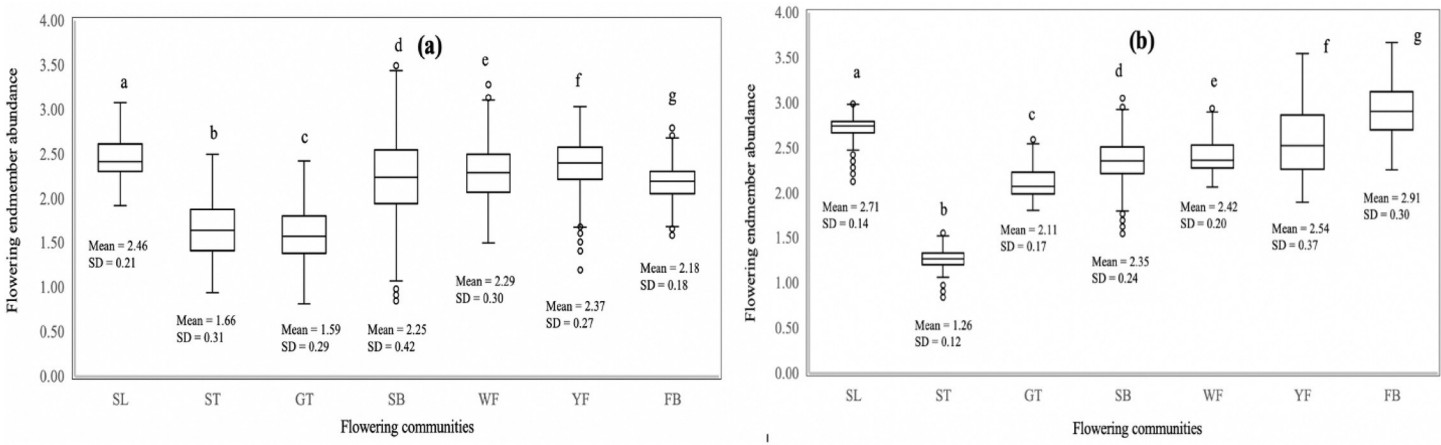

**Fig 5. Box plots distribution of the flowering endmember abundancy of the flowering vegetation communities.** Endmember standard deviation (SD) and means are shown for each of the flowering vegetation communities during the onset of flowering in January 2014 (a) and peak flowering season in February 2013 (b). Box plots with different letter(s) are significantly different from each other according to the Duncan's test ($p \leqq 0.05$). The circles show outliers.

others, flower colour [22,74–76]. Our study shows that large canopy white flowered acacia trees (predominantly *Acacia tortilis* and *Acacia brevispica*), white flowered *Terminalia brownee* and yellow flowered acacia (predominantly *Acacia nilitica*) were mainly observed on low altitude areas within the study area. These three *Acacia* spp and *Terminalia brownee* were observed to be the most foraged plants by honeybees in the Mwingi region [2]. Therefore, apiaries in such areas could be placed within the flight radius for honeybees to capitalize on such flowering plants for optimised productivity.

The high flower mapping accuracy achieved in our study could be attributed to the relatively large flowering plant canopies, which ranged between 8 and 42 feet. These flowering plants were also characterized by higher blooming density and compaction [38], particularly during the peak flowering season in February. A large plant canopy size with higher flower density enhance the floral spectral signal detected by the sensor and reduce the background heterogeneity on flowering signal, leading to improved flower detection accuracies [77].

Our results showed a better delineation of flowering plants using multispectral sensors with better spectral resolution. For instance, Sentinel-2, with relatively coarser spatial resolution (10m), mapped flowering plants with better accuracy than RapidEye (with 5 m spatial resolution) and Spot 6 (with 6 m spatial resolution). This is consistent with Mumby and Edwards (2002) who demonstrated better mapping accuracy of marine environments using higher spectral resolution images as opposed to images with fewer number of bands. Mumby and Edwards (2002) and Stratoulias et al., (2015) noted that in discriminating vegetation communities, spectral resolution could be more influential than spatial resolution.

Whereas other studies have argued that spatial resolution is more influential in vegetation mapping than spectral resolution [46,47,63], Abdel-Rahman et al. (2015) and Landmann et al. (2015) noted that spectral resolution improves the ability of an image to identify finer vegetation features like flower colour and compaction. However, we acknowledge that spatial resolution is complimentary to spectral resolution when mapping features such as flowering since flower blooms are more often smaller in size compared to other background features [78,79]. It would therefore be expected that an image with higher spatial resolution would yield higher accuracy when coupled with higher spectral resolution.

The classification of the January 2014 image revealed that the onset of the flowering season was dominated by green trees and shrubs as compared to the peak flowering season (February

2013) image. Some of the trees, such as *Combretum* spp and *Terminalia brownee*, that were identified as green at the onset of flowering season (January 2014 image) were observed to be flowering at the peak of flowering (February 2013 image). The flowering of these species is triggered by the onset of the October and April rains [80]. Although *Terminalia brownee* and *Combretum* spp scored lower than the *Acacia* spp in honey bee foraging preferences [48], their adaptation at higher altitudes provide a viable foraging option for honey bee colonies from nearby apiaries.

The flowering fobs were also observed at higher and lower altitudes. These flowering fobs (predominantly *Ipomoea kituiensis vatke*) had the highest classification accuracies compared to the other flower classes studied. This finding is consistent with Landmann et al. (2015) who established that flowering fobs were the most accurately mapped class in the study area using hyperspectral data and spectral unmixing approach. The flowering fobs were more spread across the landscape as compared to other flowers that were mainly restricted to farms and vast open grounds. The flowering fobs had higher flower compaction, which could have supressed the background interferences from adjoining green leaves and soil. In this regard, identification of their specific spectral signature was more accurate, hence the higher classification accuracy.

Flowering fobs, being ephemeral, are characterized by very short growing periods, especially on the higher grounds of the study site. The flowering fobs respond to the change in rainfall both in growth vigour and flowering intensity. The flowering fobs sprout out at the onset of the rains but lack in sustaining the rapid growth with low rainfall, hence rapidly die off. These flowering fobs however bridge the pollen and nectar gaps in low rainfall periods for improved bee diversity and honey production [2].

Additionally, there were more yellow flowers at the onset of the flowering season than at the peak of the flowering season at the study site. It was also noted that *Acacia nilotica* (with yellow flowers) bloomed earlier than the other flowering trees. These tree species (*Acacia nilitica* and *Combretum* spp) and flowering crops blossom as early as October, hence could provide foraging options to honey bees before the peak flowering periods. It's worth noting that our results showed over prediction of senesced trees during the peak flowering period, especially in images with low spatial and spectral resolutions (Spot-6 and RapidEye). This could be due to the confusion with the signature of some white flowering plants, soil, and trash, especially from bare farmlands.

Yellow flowers (predominantly *Acacia nilotica*), were generally better mapped across all the data sets than white flowers. On the other hand, white flowering trees were mapped with the least accuracy as compared to other large sized flower blooms. This is consistent with Abdel-Rahman et al. (2015), who reported that white flowers signal was confused with other classes, especially soil, leading to their lower mapping accuracy than the yellow flowers. This could explain the low deviation in January 2014 as the yellow flowers were more compact during this period as compared to the February 2013 period. Landmann et al. (2015) also established that the user accuracy of mapping white flowering plants was lower during the peak than the onset of flowering period due to confusion with dry white sand at the study site. The elevated water content in soil at the start of the flowering period helps distinguish the soil signature from white flowering plants. As opposed to the onset of flowering at the study site, peak flowering period is characterized by low rainfall and increased temperature. Therefore, the soil has bright characteristics that are similar to those of white flowering plants.

## 5. Conclusions

This study demonstrated the possibility of using multispectral images to map flowering plants in a semi-arid African savannah. The multispectral images are easily accessible, less expensive

with less complex flower map production methods. Overall, the multispectral images tested produced acceptable classification accuracies (over 67%) which improved with both spatial and spectral resolutions. WorldView-2 produced maps with the highest classification accuracy while Spot-6 had the least classification accuracy. It was also evident that spectral resolution was paramount in mapping flowering plants. Increasing spectral resolution resulted in better classification accuracies. Moreover, this study poised the freely available medium resolution Sentinel-2 as a valuable dataset for mapping flowering patterns in the African Savannahs. Additionally, the methods used herein are more practical and available to practitioners with limited remote sensing knowledge, skills and resources, hence could be used to generate more information to farmers in the semi-arid African savannahs. Results from this study could therefore be used to improve farmers' access to 'educated' information on the optimal locations for setting up apiaries in the African savannahs to maximize honeybee products output and ecological services such as pollination. Even though this research considered images from three different flowering periods, this methodology could be more applicable to images from other vegetation periods within the. This could aid in phasing the optimal times for plant specific mapping and feature selection, which could be upscaled to wider regions.

## Supporting information

**S1 Raw data. Resampled February 2013 WorldView-2 image.**
(TIF)

**S2 Raw data. Resampled February 2013 RapidEye image.**
(TIF)

**S3 Raw data. Resampled February 2013 Spot-6 image.**
(TIF)

**S4 Raw data. Resampled February 2013 Sentinel-2 image.**
(TIF)

**S5 Raw data. Resampled January 2014 WorldView-2 image.**
(TIF)

**S6 Raw data. Resampled January 2014 RapidEye image.**
(TIF)

**S7 Raw data. Resampled January 2014 Spot-6 image.**
(TIF)

**S8 Raw data. Resampled January 2014 Sentinel-2 image.**
(TIF)

**S9 Raw data. April 2014 WorldView-2 image.**
(TIF)

**S10 Raw data. 2013 field points.**
(ZIP)

**S11 Raw data. 2014 field points.**
(ZIP)

**S1 Table. Photos of representative plants of flowering functional groups used.**
(PDF)

## Acknowledgments

We are indebted to the farmers of Mwingi for their contribution towards data collection and information gathering used in this manuscript. We give our gratitude to Mutemi and Munywoki for dedicating their valuable time to take the team round even to the very remote areas of the Kasanga area in Mwingi, and for being the invaluable link with some of the local people. Also, we would like to thank Betty Sidi, Sebit Diyar, Saum Hassan, Henry Oindo, Lesley Karwitha and Davis Okoth for providing ample working conditions and ensuring all that was needed for this research was availed. The views expressed herein do not necessarily reflect the official opinion of the people and agencies that supported the authors.

## Author Contributions

**Conceptualization:** David Masereti Makori, Tobias Landmann, Onisimo Mutanga, John Odindi, Henry E. Tonnang, Suresh Raina.

**Data curation:** David Masereti Makori, Evelyn Nguku.

**Formal analysis:** David Masereti Makori.

**Funding acquisition:** Evelyn Nguku, Henry E. Tonnang, Suresh Raina.

**Investigation:** David Masereti Makori, Elfatih M. Abdel-Rahman, Tobias Landmann, John Odindi, Evelyn Nguku.

**Methodology:** David Masereti Makori, Elfatih M. Abdel-Rahman, Tobias Landmann, Onisimo Mutanga, John Odindi.

**Project administration:** Elfatih M. Abdel-Rahman, Tobias Landmann, Onisimo Mutanga, John Odindi, Evelyn Nguku, Henry E. Tonnang, Suresh Raina.

**Resources:** David Masereti Makori, Elfatih M. Abdel-Rahman, Tobias Landmann, John Odindi, Evelyn Nguku, Henry E. Tonnang, Suresh Raina.

**Software:** David Masereti Makori, Elfatih M. Abdel-Rahman, Tobias Landmann, John Odindi.

**Supervision:** David Masereti Makori, Elfatih M. Abdel-Rahman, Tobias Landmann, Onisimo Mutanga, John Odindi, Evelyn Nguku, Henry E. Tonnang, Suresh Raina.

**Validation:** David Masereti Makori, Elfatih M. Abdel-Rahman, Onisimo Mutanga, John Odindi.

**Visualization:** David Masereti Makori, Elfatih M. Abdel-Rahman, Onisimo Mutanga.

**Writing – original draft:** David Masereti Makori, Elfatih M. Abdel-Rahman.

**Writing – review & editing:** David Masereti Makori, Elfatih M. Abdel-Rahman, Tobias Landmann, Onisimo Mutanga, John Odindi, Evelyn Nguku, Henry E. Tonnang, Suresh Raina.

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
