## [Decision Letter · Decision Letter 0]

21 May 2020

PONE-D-20-10319

Suitability of resampled multispectral datasets for mapping flowering plants in the Kenyan savannah

PLOS ONE

Dear David Masereti Makori,

Thank you for submitting your manuscript to PLOS ONE. After careful consideration, we feel that it has merit but does not fully meet PLOS ONE’s publication criteria as it currently stands. Therefore, we invite you to submit a revised version of the manuscript that addresses the points raised during the review process.

We have received comments from two reviewers who have provided their concerns on the paper and suggestions to improve it. Please find the comments at the end of this email.

We would appreciate receiving your revised manuscript by 30th June. To enhance the reproducibility of your results, we recommend that if applicable you deposit your laboratory protocols in protocols.io, where a protocol can be assigned its own identifier (DOI) such that it can be cited independently in the future. For instructions see: http://journals.plos.org/plosone/s/submission-guidelines#loc-laboratory-protocols

We look forward to receiving your revised manuscript.

Kind regards,

Wang Li

Academic Editor

PLOS ONE

Journal Requirements:

2.We suggest you thoroughly copyedit your manuscript for language usage, spelling, and grammar. If you do not know anyone who can help you do this, you may wish to consider employing a professional scientific editing service.  

'We gratefully acknowledge the financial support for this research by the following organizations and agencies; The European Union, project FED/2013/319-933; the Centre for International Migration and Development (CIM) of the German Development Agency (GIZ); UK’s Department for International Development (DFID); Swedish International Development Cooperation Agency (SIDA); the Swiss Agency for Development and Cooperation (SDC); and the Kenyan Government....our sincere gratitude goes to the European Union for funding this project (Project number: DCI-FOOD-2011/023-520)....'

'SR;

FED/2013/319-933 and DCI-FOOD-2011/023-520;

Europan Union;

https://europa.eu/;

NO.'

Please clarify the sources of funding (financial or material support) for your study. List in full the grants or organizations that supported your study, including funding received from your institution.State what role the funders took in the study. If the funders had no role in your study, please state: “The funders had no role in study design, data collection and analysis, decision to publish, or preparation of the manuscript.”If any authors received a salary from any of your funders, please state which authors and which funders.If you did not receive any funding for this study, please state: “The authors received no specific funding for this work.”

4. We note that Figures 1-4 in your submission contain map/satellite images which may be copyrighted.

a. You may seek permission from the original copyright holder of Figures 1-4 to publish the content specifically under the CC BY 4.0 license. 

5. Please upload a copy of Figures 6 and 7, to which you refer in your text. If either figure is no longer to be included as part of the submission please remove all reference to it within the text.

Reviewers' comments:

Reviewer's Responses to Questions

**Comments to the Author**

1. Is the manuscript technically sound, and do the data support the conclusions?

Reviewer #1: Partly

Reviewer #2: Yes

2. Has the statistical analysis been performed appropriately and rigorously? 

Reviewer #1: N/A

Reviewer #2: Yes

3. Have the authors made all data underlying the findings in their manuscript fully available?

Reviewer #1: No

Reviewer #2: No

4. Is the manuscript presented in an intelligible fashion and written in standard English?

Reviewer #1: Yes

Reviewer #2: Yes

5. Review Comments to the Author

Reviewer #1: Dear editor,

I have reviewed the paper entitled “Suitability of resampled multispectral datasets for mapping flowering plants in the Kenyan savannah”. The topic of the paper is interesting as the authors tries to use the flower color to identify tree species, it is creative; but the paper is more likely a project report as the scientific question is not clear. I think the authors should re-organize the paper before I can recommend it for publication.

Here are some major concerns:

1. What is the shortage of the current plant species classification? This question is not clearly addressed in the introduction section. This paper should focus on proposing new methodologies for plant species classification (maybe by using flower color, and flower color could be detected by earth observations).

2. I cannot understand why the AISA Eagle hyperspectral images were simulated as Worldview, Sentinel-2, SPOT-6. Particularly, the Sentinel-2 data is free available currently. And I think the comparison among the four simulated sensor images in this way make no sense. If the worldview, sentinel-2, spot-6 is not available, using hyperspectral images for flower color and plant species mapping is also fantasticating.

3. The paper just used two images in Jan 2014 and Feb 2013 for flower color mapping, but is this flowering time suitable for a large-scale mapping? I suggest to acquire satellite image of the whole year round and analyze the optimal time phase for plant specifies mapping, this optimal-feature-selection is a common procedure when trying to apply the conclusion of this paper to other study regions. Authors should read more papers about optimal feature selection for this issue.

Reviewer #2: Well written paper. However, the paper has adopted a heuristic approach rather a hypothesis based approach. If you would ask me, I am not at all surprised by the results obtained. The power of Worldview-2 in land targets classification has been assessed in many recent studies. Therefore, the problem statement is not strong. The novelty could be in the application - for mapping flowering plants important for pollinator survival. Therefore, much more biological information of the flowering plants would help strengthen this aspect e.g. table of pollinators in the region and feeding patterns could justify the mapping exercise, because so far as I am concern, there is not novelty in the remote sensing exercise. You could then analyse the impact of the distribution of flowering plants on the pollinator populations.

Minor issues

- why would you consider Rapideye, Spot 6, Worldview-2 and sentinel-2 as high spectral resolution sensors?

- RF in a classification mode, it is growing regression trees - as mentioned in the paper? What is the exact algorithm used in the training of each tree?

- It is not sufficient to simply mention that resampling tool in envi was used. Which algorithm and why. The best would be to use the sensor's response function.

- Actual pictures of the flowering communities could be included.

6. PLOS authors have the option to publish the peer review history of their article (what does this mean?). If published, this will include your full peer review and any attached files.

Reviewer #1: No

Reviewer #2: No

---

## [Author Response · Author response to Decision Letter 0]

7 Jul 2020

Response to Reviewers

We have responded to the reviewers’ comments by highlighting our comments in GREEN, RED and BLUE for comments shared by both reviewer #1 and #2, reviewer #1, and reviewer #2, respectively. To make tracing our changes in the document easy by the reviewers, we have provided the revised manuscript with track changes for ease of identification.

Please find the corrections for reviewer#1 and reviewer#2 below. 

Journal Requirements

Concerns addressed under PONE-D-20-10319

Authors’ response

We have edited the manuscript for consistency and ensured that it meets PLOS ONE’s style requirements, including file naming. 

Authors’ response

We have gone through the manuscript multiple times to ensure that language usage, spelling and grammar meet PLOS ONE standards. 

'We gratefully acknowledge the financial support for this research by the following organizations and agencies; The European Union, project FED/2013/319-933; the Centre for International Migration and Development (CIM) of the German Development Agency (GIZ); UK’s Department for International Development (DFID); Swedish International Development Cooperation Agency (SIDA); the Swiss Agency for Development and Cooperation (SDC); and the Kenyan Government....our sincere gratitude goes to the European Union for funding this project (Project number: DCI-FOOD-2011/023-520)....'

'SR;

FED/2013/319-933 and DCI-FOOD-2011/023-520;

European Union;

https://europa.eu/;

NO.'

a. Please clarify the sources of funding (financial or material support) for your study. List in full the grants or organizations that supported your study, including funding received from your institution.

d. If you did not receive any funding for this study, please state: “The authors received no specific funding for this work.”

Authors’ response

We have edited and removed the funding related text from the acknowledgement section of the manuscript. We further would want to amend our funding statement to read as follows;

The authors are grateful to the European Union for funding the project “African reference laboratory (with satellite stations) for the management of pollinator bee diseases and pests for food security” under grant number DCI-FOOD-2011/023-520, in which context the current study was made possible. In addition, the authors thank the European Union for facilitating the acquisition of the April 2014 WorldView-2 data that was used in this study. Further, the authors are sincerely grateful to the University of Helsinki funding research community GIMMEC–Geoinformatics for monitoring and modelling of environmental change for facilitating the acquisition and processing of the AISA Eagle hyperspectral data. David M. Makori, Evelyn Nguku, Suresh Raina and Tobias Landmann salary’s were made possible by the financial support given by the European Union through project DCI-FOOD-2011/023-520, while Elfatih M. Abdel-Rahman, Evelyn Nguku, Henry E. Z. Tonnang and Tobias Landmann received salaries from icipe. We also gratefully acknowledge the icipe core funding provided by UK Aid from the Government of the United Kingdom; Department for International Development Cooperation Agency (Sida); the Swiss Agency for Development for Economic Cooperation and Development (BMZ), Germany Federal Democratic Republic of Ethiopia; and the Kenyan Government. The funders had no role in study design, data collection and analysis, decision to publish, or preparation of the manuscript. 

4. We note that Figures 1-4 in your submission contain map/satellite images which may be copyrighted.

We require you to either (a) present written permission from the copyright holder to publish these figures specifically under the CC BY 4.0 license, or (b) remove the figures from your submission

Authors’ response

The hyperspectral and the 10cm Nikon datasets utilized in this research were obtained by flying an AISA Eagle sensor and the Nikon camera over our Mwingi Study site in both 2013 and 2014. This was done by ICIPE in collaboration with the University of Helsinki, Department of Geoscience and Geography under the University of Helsinki funding research community GIMMEC–Geoinformatics for monitoring and modelling of environmental change. The multispectral images were resampled from the two hyperspectral images to yield products within the same year. The April 2014 Worldview-2 image was bought by ICIPE with facilitation of the European Union’s funded Bee Health Project for this research, subsequently making the image data the property of ICIPE. 

5. Please upload a copy of Figures 6 and 7, to which you refer in your text. If either figure is no longer to be included as part of the submission please remove all reference to it within the text.

Authors’ response

Reference to Figures 6 and 7 in the text in the manuscript was erroneously included. We have proceeded to remove the references from the text accordingly. 

Concerns addressed under PONE-D-20-10319R1

1) Thank you for updating your data availability statement. You note that your data are available within the Supporting Information files, but no such files have been included with your submission. At this time we ask that you please upload your minimal data set as a Supporting Information file, or to a public repository such as Figshare or Dryad.

Please also ensure that when you upload your file you include separate captions for your supplementary files at the end of your manuscript.

As soon as you confirm the location of the data underlying your findings, we will be able to proceed with the review of your submission.

Authors’ response

Thank you for the kind reminder. We have proceeded to add supporting information and have included separate captions under the Supporting information of the manuscript at the end. These are listed below;

i. S1 Raw data. Resampled February 2013 WorldView-2 image (tif).

ii. S2 Raw data. Resampled February 2013 RapidEye image (tif).

iii. S3 Raw data. Resampled February 2013 Spot-6 image (tif).

iv. S4 Raw data. Resampled February 2013 Sentinel-2 image (tif).

v. S5 Raw data. Resampled January 2014 WorldView-2 image (tif).

vi. S6 Raw data. Resampled January 2014 RapidEye image (tif).

vii. S7 Raw data. Resampled January 2014 Spot-6 image (tif).

viii. S8 Raw data. Resampled January 2014 Sentinel-2 image (tif).

ix. S9 Raw data. April 2014 WorldView-2 image (tif).

x. S10 Raw data. 2013 field points (shp).

xi. S11 Raw data. 2014 field points (shp).

xii. S12 Content Permission Form by ICIPE (pdf).

2) Thank you for providing further details regarding the creation and ownership of the images that appear in this manuscript. Before we can proceed, however, we ask that you provide additional details regarding the following figures and tables.

a) TABLE 6: Please indicate where the authors obtained the 7 images of trees, crops, and flowers, or if the authors took the photographs themselves.

Authors’ response

We have edited the caption of Table 6 to include information on the acquisition of the 7 images of trees, crops and flowers. We have included this information in the caption;

“These photos were taken by the lead author, David Makori during field surveys curried out in January 2014 and February 2013.”

b) FIGURE 1-4: Since the image data are the property of ICIPE, please indicate what license they've been copyrighted under.

- Please provide links to all of the image data retrieved from ICIPE for these figures, if possible

- If the copyright license is more restrictive than CC BY 4.0, please note that you will need to request permission from the copyright holder to publish these images in PLOS ONE under a CC BY 4.0 license.

To seek permission from the copyright owner to publish these figures under the Creative Commons Attribution License (CCAL), CC BY 4.0, please contact them with the following text and PLOS ONE Request for Permission form (http://journals.plos.org/plosone/s/file?id=7c09/content-permission-form.pdf):

“I request permission for the open-access journal PLOS ONE to publish XXX under the Creative Commons Attribution License (CCAL) CC BY 4.0 (http://creativecommons.org/licenses/by/4.0/). Please be aware that this license allows unrestricted use and distribution, even commercially, by third parties. Please reply and provide explicit written permission to publish XXX under a CC BY license.”

Please upload the granted permission to the manuscript as a Supporting Information file. In the figure caption of the copyrighted figure, please include the following text: “Republished from [ref] under a CC BY license, with permission from [name of publisher], original copyright [original copyright year].”

Please note that RightsLink permission forms often impose use restrictions that are incompatible with our CC BY 4.0 license, and we are therefore unable to accept these permissions. For this reason, we strongly recommend contacting copyright holders with the PLOS ONE Request for Permission form.

If you are unable to obtain permission from the original copyright holder, please either remove the figure or supply a replacement figure that complies with the CC BY 4.0 license. Please check copyright information on all replacement figures and update the figure caption with source information. If applicable, please specify in the figure caption text when a figure is similar but not identical to the original image used in the study, and is therefore for illustrative purposes only.

To seek permission from the copyright owner to publish these figures under the Creative Commons Attribution License (CCAL), CC BY 4.0, please contact them with the following text and PLOS ONE Request for Permission form (http://journals.plos.org/plosone/s/file?id=7c09/content-permission-form.pdf):

Authors’ response

We are very grateful for guiding us on how to provide the right information and permission from the copyright holder. The Director General and CEO of ICIPE, the owner of the images (AISA Eagle and WorldView-2) used in this study filled the Content Permission Form which has been subsequently attached and referenced under the Supporting Information section under S12 Content Permission Form by ICIPE (pdf). 

c) FIGURE 1: Please also provide source information for the shape files and basemaps used to create these images, and include links. Indicate as well what software was used to create the figure.

Authors’ response

We have provided information on the source of the shape files and basemaps used in Figure 1 and included links in the caption. We have further indicated the software used in the generation of this map, provided links to access the software and the necessary reference information. 

d) FIGURE 2-4: Please describe what software was used to create these figures, and indicate whether any additional shapefiles etc. were used in creating it.

Authors’ response

We have indicated the software used to generate these maps in the respective captions. We have further provided the link to access the software and thee reference to the software for each of the figures. 

 

Reviewer's Responses to Journal Questions

1. Is the manuscript technically sound, and do the data support the conclusions?

Reviewer #1: Partly

Reviewer #2: Yes

Authors’ response

We thank the reviewers for their input to our manuscript. We are also grateful to Reviewer#2 for confirmation that the manuscript is technically sound and that the data supports the conclusion. We appreciate Reviewer#1’s comment that we have taken time to address some of the areas of the manuscript that were not satisfactorily written. We have revised the introduction and conclusion to support the results in the manuscript. 

2. Has the statistical analysis been performed appropriately and rigorously?

Reviewer #1: N/A

Reviewer #2: Yes

Authors’ response

We thank the reviewers for their input on the statistical soundness of our manuscript. The concerns raised have been addressed and are discussed under relevant sections. 

3. Have the authors made all data underlying the findings in their manuscript fully available?

Reviewer #1: No

Reviewer #2: No

Authors’ response

We are grateful for this response. We have now made the data used for analysis in this manuscript available. The data availed include;

i. Resampled images for both February 2013 and January 2014 for the following sensors

a. WorldView-2

b. RapidEye

c. Spot-6

d. Sentinel-2

ii. April 2014 WorldView-2 original data

iii. Field points for both 2013 and 2014.

4. Is the manuscript presented in an intelligible fashion and written in standard English?

Reviewer #1: Yes

Reviewer #2: Yes

Authors’ response

We thank the reviewers for finding this manuscript to be presented in an intelligible fashion and written in standard English. 

 

Reviewers’ Comments

Reviewer#1

I have reviewed the paper entitled “Suitability of resampled multispectral datasets for mapping flowering plants in the Kenyan savannah”. The topic of the paper is interesting as the authors tries to use the flower color to identify tree species, it is creative; but the paper is more likely a project report as the scientific question is not clear. I think the authors should re-organize the paper before I can recommend it for publication.

Authors’ response

We thank the reviewer for taking time to go through the manuscript and finding it interesting and creative. We have addressed the comments raised herein and reorganized the manuscript in key areas to meet the standard necessary for publication in PLOS ONE. We have reorganized the introduction and information on the importance of the pollinators and flowering plants to the ecosystem and the surrounding community. In addition, the use of multispectral images to map functional flowering plant groups using flower colour has been reinforced. We have added the following text in the introduction of the manuscript - line 26 to 63; 

‘African savannahs are characterized by unreliable and erratic rainfall with low and dispersed forest pockets. They do not efficiently support rain-fed agriculture, necessitating for alternative sources of income to supplement the unpredictable crop yield (1,2). Apiculture and related ecosystem services such as pollinator activities boosts local economies, food and nutritional security and improve biodiversity, hence valuable and sustainable socio-ecological practices in the African savannah (3,4).

Insects such as honeybees, stingless bees, wasps, butterflies, mulberry and wild silk moths (2,5) are important income sources to local communities and paramount in pollination and forest conservation. These insects are valued for their production of honey, wax, dyes and silk on one hand (6,7), and pollination of agricultural and horticultural crops, forest ecosystems, as well as conservation of derelict land and degraded forest (2,8). It is estimated that between 60% to 90% of plant species depend on insects and other animals such as birds for pollination (2,9). In the USA for instance, pollination is estimated to contribute more than $14 billions a year to the agricultural economy alone (4,10). However, whereas the contribution of pollination in Africa is unequivocal, its economic importance is yet to be fully documented. 

As human pressure on land increases, communities living within four kilometers radius from the forest increases. These communities directly or indirectly depend on a range of forest resources and ecosystem services (3,4). For instance, placement of apiaries in close proximity of a forest (less than one kilometer) doubles the production of honey than when placed out of a three kilometer radius (11). Moreover, proximity to the right types and amounts of pollen and nectar improve hive productivity, honey quality and the agility of bees to fight off pests and diseases (5,11,12). Forest habitats are important to the various life cycles of many beneficial insect species (13,14). These insects are useful for among others pollination, improving biodiversity, diversification of livelihood options and natural agents of pest control (15,16). Disturbance of forest stands and savannah vegetation leads to reduction in pollen and nectar sources and ultimately pollinators which are susceptible to habitat alteration and changes in climate (17). A decline or elimination of some plant species leads to a reduction of pollinators specific to either pollen type, flower colour and morphology or physiology (18). Most insects rely on visual signals in the choice of flowers to visit. Colour, shape and size influence insects in flower preference (19–21). For instance, hummingbirds prefer red coloured flowers, flies like pale colours, while butterflies and bees prefer brightly coloured flowers (19–23). Since most of the crops are pollinated by social pollinators such as honeybees, agricultural production in Africa is predicted to reduce as their numbers decline (24,25). In this regard, measures aimed at locating, conserving and improving cover of relevant flowering vegetation around vulnerable communities that depend on agriculture for their livelihoods are necessary.’

Major concerns:

i. What is the shortage of the current plant species classification? This question is not clearly addressed in the introduction section. This paper should focus on proposing new methodologies for plant species classification (maybe by using flower color, and flower color could be detected by earth observations).

Authors’ response

We thank the reviewer for this question. To the best of our knowledge, this study is unique as multispectral images have not been used to map functional flowering groups of melliferous plants. We have therefore included in the manuscript the uniqueness of this study in the introduction (line 69), thus;

‘operational mapping of tree species using remote sensing systems is hampered by the low spectral resolution in multispectral images and high acquisition cost of hyperspectral images’

And further in line 78;

‘it is paramount to explore the utility of multispectral images with fewer broad bands for optimal discrimination of functional flowering groups’.

ii. I cannot understand why the AISA Eagle hyperspectral images were simulated as Worldview, Sentinel-2, SPOT-6. Particularly, the Sentinel-2 data is free available currently. And I think the comparison among the four simulated sensor images in this way make no sense. If the worldview, sentinel-2, spot-6 is not available, using hyperspectral images for flower color and plant species mapping is also fantasticating.

Authors’ response

We thank the reviewer for this comment. The acquisition cost of most multispectral images used in this study were very high during the time of the study. Hence, we opted to firstly test their utility for mapping flowering plants, since we already had the AISA Eagle hyperspectral images that we resampled to the multispectral sensors. We have highlighted the cost of acquiring each of the image in the methodology section and we have further stated this in the introduction section in line 88, thus;

‘the acquisition cost of some of the commercial multispectral data could limit their operational mapping applications. Hence, there is need first to explore the utility of simulated image data of such multispectral sensors and compare their usefulness with freely available ones for flower mapping’. 

Even though Sentinel-2 is currently freely available, it was not available by the time this study was being carried out. Sentinel-2 was launched in 2015 while this study considered images from February 2013 and January 2014. Therefore, we could not acquire images from this satellite to use for this study, hence the resampling from AISA Eagle. We further indicated this in the manuscript in line 252;

‘When this study was conducted, the sensor was not yet launched hence the need to resample this dataset from AISA Eagle images.’

Further, the use hyperspectral data to map flowering plants has been explored before by Landmann et. al (2015) and Abdel-Rahman et. al (2015), but as afore mentioned above and in the manuscript in line 88, this data is expensive and analysis methods used are very complex, making them unavailable to many who desire to use these methods for improved farming, necessitating the consideration of our methods. 

iii. The paper just used two images in Jan 2014 and Feb 2013 for flower color mapping, but is this flowering time suitable for a large-scale mapping? I suggest to acquire satellite image of the whole year round and analyze the optimal time phase for plant specifies mapping, this optimal-feature-selection is a common procedure when trying to apply the conclusion of this paper to other study regions. Authors should read more papers about optimal feature selection for this issue.

Authors’ response

We thank the reviewer for this comment, and we took a kind note of this concern. The flowering periods for most plants in the study area have been established to be between December to May each year, with the peak in February. This study acquired images for time periods January; onset of flowering, February; peak flowering and April; end of flowering, so as to maximize the flowering season. Outside this season, most plants are either in senescence or green and would not provide the much-needed information for this study on flower colour. In addition, acquiring some of these datasets is very expensive and may not be feasible to assemble images covering the whole year. We have however recommended in our conclusion that our methods could be strengthened by using easily available images to make a year-round map that could be upscaled, thus in line 479;

‘Even though this research considered images from three different flowering periods, this methodology could be more applicable and upgradable by incorporating images from other vegetation periods within the year that may not necessarily be flowering. This could aid in phasing the optimal times for plant specific mapping and feature selection, which could be upscaled to wider regions.’

Reviewer #2: 

Well written paper. However, the paper has adopted a heuristic approach rather a hypothesis based approach. If you would ask me, I am not at all surprised by the results obtained. The power of Worldview-2 in land targets classification has been assessed in many recent studies. Therefore, the problem statement is not strong. The novelty could be in the application - for mapping flowering plants important for pollinator survival. Therefore, much more biological information of the flowering plants would help strengthen this aspect e.g. table of pollinators in the region and feeding patterns could justify the mapping exercise, because so far as I am concern, there is not novelty in the remote sensing exercise. You could then analyse the impact of the distribution of flowering plants on the pollinator populations.

Authors’ response

We thank the reviewer for the critical criticism. We are also glad that the reviewer agrees concurs on the importance of mapping flowering plants for the survival of pollinators, improving the livelihoods of local communities and as an incentive to conservation efforts. Also, reviewer#1 raised up the same concern. To strengthen the problem statement, we have included biological information on the pollinators, flowering plants and how they link to the communities living around the forest and impact forest conservation. This has been linked to the need to map and locate these flowering plants using multispectral images and methods that are easily accessible to rural communities living in the African savannahs, to inform the placement of apiaries and improve pollination on their crops. Kindly see lines 26 to 62.

Minor issues

i. why would you consider Rapideye, Spot 6, Worldview-2 and sentinel-2 as high spectral resolution sensors?

Authors’ response

We realize that the context in which this statement was written was not clear; ‘The newly launched higher spectral and/or spatial resolution sensors such as WorldView-2, RapidEye, Spot-6 and Sentinel-2’. We have therefore replaced it with this statement in line 83; ‘The newly launched relatively improved spectral and/or spatial resolution sensors such as WorldView-2, RapidEye, Spot-6 and Sentinel-2’. We hope the corrected statement will clearly pass the intended message, that is, the four multispectral images have improved spectral and spatial resolution than other multispectral images, for instance Landsat TM. 

ii. RF in a classification mode, it is growing regression trees - as mentioned in the paper? What is the exact algorithm used in the training of each tree?

Authors’ response

We thank the reviewer for this comment. We have added information on the RF classification in the manuscript in line 221-239, which describes the algorithm used for training of each tree. 

iii. It is not sufficient to simply mention that resampling tool in envi was used. Which algorithm and why. The best would be to use the sensor's response function.

Authors’ response

We note the concerns of the reviewer on the resampling tool used in ENVI. We have subsequently included detailed information on the method used for resampling in ENVI under the resampling tool using the predefined filter function - line 139 onwards. This could be easily understood by the readers and easily upscalable by those interested. 

iv. Actual pictures of the flowering communities could be included.

Authors’ response

We thank the reviewer for this suggestion. Actual pictures of the different flowering plants in the functional groups under consideration in this study have been included in Table 6 from line 224.

---

## [Decision Letter · Decision Letter 1]

6 Aug 2020

Suitability of resampled multispectral datasets for mapping flowering plants in the Kenyan savannah

PONE-D-20-10319R1

Dear Dr. Makori,

We’re pleased to inform you that your manuscript has been judged scientifically suitable for publication and will be formally accepted for publication once it meets all outstanding technical requirements.

Kind regards,

Wang Li

Academic Editor

PLOS ONE

Additional Editor Comments (optional):

Reviewers' comments:

Reviewer's Responses to Questions

**Comments to the Author**

1. If the authors have adequately addressed your comments raised in a previous round of review and you feel that this manuscript is now acceptable for publication, you may indicate that here to bypass the “Comments to the Author” section, enter your conflict of interest statement in the “Confidential to Editor” section, and submit your "Accept" recommendation.

Reviewer #1: All comments have been addressed

Reviewer #2: All comments have been addressed

2. Is the manuscript technically sound, and do the data support the conclusions?

Reviewer #1: Yes

Reviewer #2: Yes

3. Has the statistical analysis been performed appropriately and rigorously? 

Reviewer #1: Yes

Reviewer #2: Yes

4. Have the authors made all data underlying the findings in their manuscript fully available?

Reviewer #1: Yes

Reviewer #2: Yes

5. Is the manuscript presented in an intelligible fashion and written in standard English?

Reviewer #1: Yes

Reviewer #2: Yes

6. Review Comments to the Author

Reviewer #1: Dear editor, I have reviewed the revised paper, and all my concerns have been addressed. I am satisfied with the revision.

Reviewer #2: Thanks. The authors have adequately address my comments. The study demonstrates the use of remote sensing for mapping flowering plants and periods an a typical African savanna using hyperspectral data resampled to several multispectral data. The results offer promise for beehive management and honey production in these systems.

7. PLOS authors have the option to publish the peer review history of their article (what does this mean?). If published, this will include your full peer review and any attached files.

Reviewer #1: No

Reviewer #2: No

---

## [Editor Report · Acceptance letter]

10 Aug 2020

PONE-D-20-10319R1 

Suitability of resampled multispectral datasets for mapping flowering plants in the Kenyan savannah 

Dear Dr. Makori:

I'm pleased to inform you that your manuscript has been deemed suitable for publication in PLOS ONE. Congratulations! Your manuscript is now with our production department. 

Kind regards, 

on behalf of

Dr. Wang Li 

Academic Editor

PLOS ONE